# Evaluation of Functional Quality of Maize with Different Grain Colors and Differences in Enzymatic Properties of Anthocyanin Metabolism

**DOI:** 10.3390/foods14040544

**Published:** 2025-02-07

**Authors:** Jing Li, Zhanqiang Chen, Baojie Su, Yanan Zhang, Zhiping Wang, Ke Ma, Boyu Lu, Jianhong Ren, Jianfu Xue

**Affiliations:** 1College of Agriculture, Shanxi Agricultural University, Taigu, Jinzhong 030800, China; lj332154088@163.com (J.L.); chenzhanqiang01@126.com (Z.C.); s20222137@stu.sxau.edu.cn (B.S.); z20223121@stu.sxau.edu.cn (Y.Z.); z20213112@163.com (Z.W.); b20211021@stu.sxau.edu.cn (B.L.); 2Taigu District Party School of Jinzhong City of the Communist Party of China, Taigu, Jinzhong 030800, China; 3College of Agronomy and Biotechnology, China Agricultural University, Beijing 100193, China; b20223010009@cau.edu.cn; 4College of Life Sciences, Shanxi Agricultural University, Taigu, Jinzhong 030800, China; renjh@sxau.edu.cn

**Keywords:** waxy maize, color, anthocyanin biosynthesis metabolism, functional quality, comprehensive evaluation

## Abstract

Waxy maize (*Zea mays* L. *sinensis kulesh*) contains a lot of nutrients, and has a long history of cultivation and extensive consumption. In this study, six waxy maize varieties of white (J18 and W2000), yellow (J41 and J7), and black (J10 and J20) were selected as experimental materials, and the functional nutrients and the differences in anthocyanin anabolic pathways in maize kernels at 14, 18, 22, and 26 days after pollination were determined. The result show that the varieties and kernel development stages had significant effect on the carotenoid, soluble sugar, vitamin C, anthocyanin, and mineral element content. The black waxy maize varieties had a higher anthocyanin content, which plays an important role in maize kernel coloration, whereas the yellow and black waxy maize varieties exhibit a greater abundance of mineral elements. Furthermore, the phenylalanine content, as well as the activities of phenylalanine ammonia lyase (PAL), chalcone isomerase (CHI), dihydroflavonol reductase (DFR), and flavonoid 3-glucosyltransferase (UFGT), played a significant role in the anthocyanin biosynthetic pathway. In conclusion, the comprehensive functional quality of waxy maize decreased with the delay of kernel development stage, and the black waxy maize varieties demonstrated superior functional quality. The PAL and CHI played a primary role in the initial phase of anthocyanin accumulation, while UFGT gradually assumed control in the subsequent stages.

## 1. Introduction

Maize (*Zea mays* L.) is an ancient cultivated plant, and is one of the most important cereal crops in terms of world’s staple foods [1,2]. It is commonly used in the feed and food industry, but also for other different industrial purposes such as the pharmaceutical and cosmetic fields [3]. On an estimated 197 million hectares of land worldwide, maize is grown for dry grain each year, yielding 1137 million tons (M t) of dry grain annually. The amount of maize consumed annually by humans worldwide is 18.5 kg per capita, or 11% of the average of 175 kg of grain consumed annually worldwide [4]. Among the maize classifications, waxy maize is a distinctive type cultivated in China, which boasts a widespread consumption globally [5]. Waxy maize contains dietary fiber and bioactive compounds, which is a good source of natural antioxidants. It might confer health-protective benefits by alleviating oxidative stress by preventing free radicals from damaging proteins, DNA, and lipids [6]. Therefore, it is more and more interesting to consumers as a natural food additive and dietary supplement [7]. The formation of color in maize kernels is a crucial aspect of their development [8]. The waxy maize exhibits a wide range of kernel colors, from white to dark black, with yellow kernels representing the most prevalent type cultivated globally [1,9].

The waxy maize kernels are rich in proteins, oils, vitamins, phenolic compounds, phytosterols, vitamins, minerals, and other beneficial substances [10]. Notably, colored waxy maize has garnered attention for its potential role in diet-related chronic diseases [11]. Therefore, the colored waxy maize has been favored by more and more people in recent years. Carotenoids, such as lutein and zeaxanthin, which are abundant in yellow waxy maize varieties, can help people who need vitamin A in areas where there is a shortage of fruits, vegetables, and animal products [12,13,14]. Carbohydrates are vital in waxy maize quality, and the sweetness of waxy maize is closely related to soluble sugar content [15]. However, it is still unclear whether there is a correlation between maize kernel color and soluble sugar content. Vitamin C is an essential dietary nutrient, which cannot be synthesized directly by humans [16]. Shan et al. [17] reported that the concentration of vitamin C varied among different kernel colors (purple, yellow, white) of waxy maize. Anthocyanin is a class of water-soluble flavonoids that function as antioxidants, playing a crucial role in the scavenging of free radicals [18]. Maize is a plentiful source of anthocyanin, which are found in the pericarp, the aleurone, and in both of these structures [19]. Previous studies have shown that the anthocyanin content of dark (black, purple, blue) maize is higher [20]. The presence of certain minerals is indispensable for the survival of plants and the sustenance of humans [21]. For humans, the absorption of minerals from food, including plants, is vital for growth and survival. Nevertheless, mineral malnutrition affects over two billion people globally [22,23]. Previous studies have shown that different maize varieties have significant differences in mineral absorption, which affects the mineral content of the kernel [21]. In addition, during the development of the kernel, the nutrient content of waxy maize has undergone a series of changes [17]. Yang et al. [24] reported that with the development of maize kernels, starch content gradually increased and protein content gradually decreased. Lim et al. [21] found that waxy maize had higher levels of folic acid and vitamin C in the mid and late stages of development before the drying period. Nevertheless, the distinctions in functional nutrients among waxy maize with varying kernel colors at different developmental stages remain unclear.

There has been a surge in research due to the growing importance of anthocyanin in food, medicine, health products, cosmetics, and other fields and their potential health benefits in recent years [25]. In maize, the biosynthesis of anthocyanin reflects a complex genetic interplay that results in a variety of pigmentation and phenotypic characteristics [11]. The process of anthocyanin biosynthesis typically progresses through three distinct stages [26]. Phenylalanine is the precursor for anthocyanin synthesis. Phenylana-nine lyase (PAL) catalyzes the phenylalanine in the first step of anthocyanin synthesis to produce cinnamic acid. Cinnamate-4-hydroxylase (C4H) and 4-coumarate: CoA ligase (4CL) next catalyze the formation of 4-Coumaroyl-CoA [27]. Thereinto, PAL is the first enzymatic step in anthocyanin synthesis and the initial enzyme for the synthesis of flavonoid and lignin synthesis [28]. Chalcone synthase (CHS) and chalcone isomerase (CHI) then catalyze the production of naringenin in the second stage of biosynthesis. Flavanone 3-hydroxylase (F3H) might then transform it into di-hydrokaempferol. When flavanone 3′-5′-hydroxylase (F3′5′H) and flavonoid 3′-hydroxylase (F3′H) were present, dihydroquercetin and dihydromyri-cetin were produced, respectively [27]. CHI is the second key enzyme in anthocyanin synthesis and expression of the CHI gene has an effect on anthocyanin synthesis [29]. In the third step of anthocyanin synthesis, F3H, F3′5′H and F3′ were catalyzed by dihydroflavonol-4-reductase (DFR) and anthocyanin synthetase (ANS) to form anthocyanin. Following the synthesis of anthocyanin, the colored anthocyanin are catalyzed by ANS, and subsequently by flavonoid 3-glucosyltransferase (UFGT). UFGT represents the final key enzyme in anthocyanin synthesis, facilitating the conversion of unstable anthocyanin into stable anthocyanin [30]. In the anthocyanin biosynthesis pathway, the change in enzyme activity is closely related to maize varieties, and directly determines the final anthocyanin content of grains [27,31]. Cui et al. [32] reported that dark maize had higher PAL activity, and reached the highest value 40 days after pollination, and then the activity gradually decreased. CHI and DFR expressed at higher activity during the maturity stage compared with fresh-eating stage [33]. However, further investigation is required to elucidate the limiting factors and key metabolic periods of anthocyanin synthesis in waxy maize with different kernel colors.

The aims of this study were: (i) to clarify the difference in functional substance content in waxy maize with different kernel colors; (ii) to comprehensively evaluate the functional quality of waxy maize with different kernel colors at different stages after pollination; and (iii) to investigate the limiting factors and key metabolic periods of anthocyanin synthesis in waxy maize with different kernel colors.

## 2. Materials and Methods

### 2.1. Experimental Site Description

The experiment was conducted at the Maize Experimental Base (112°36′ E, 37°58′ N), Shanxi Province in 2019. The climate of the Maize Experimental Base is classified as a temperate continental climate, exhibiting four distinct seasons. The annual average sunshine is 2577.5 h, with the daily light rate reaching 58%. The annual average temperature varies with the terrain, with an annual average precipitation of 462 mm. The frost-free period is 183 days. The previous crop in the experimental field is maize, and the nutrient content of the soil is shown in Appendix A.

### 2.2. Material

The white maize varieties Jinnuo 18 (J18) and Wannuo 2000 (W2000), yellow maize varieties Jindannuo 41 (J41) and Jinong 7 (J7), black maize varieties Jinnuo 10 (J10) and Jinnuo 20 (J20) were provided by the Shanxi Agricultural University Maize Institute. The reagents used in the trial were purchased from Sinopharm and Shanghai Suolaibao Biotechnology Co., Ltd., Shanghai, China.

### 2.3. Experimental Design

Field tests were conducted using a three-replicate randomized complete block design with a 50 m^2^ plot size. Prior to planting, the ground was prepared, and 600 kg/hm^2^ of compound fertilizer (N: P: K = 23:12:5) was administered as a base fertilizer. During the jointing period, urea was added at a rate of 337.5 kg/hm^2^. The maize was sown on May 7 and harvested on August 21 (J41, J7, and J10) and August 29 (J18, W2000, and J20), with a density of 5.25 × 10^4^ plants per ha. Diseases and insect pests were well managed at the experimental plot.

### 2.4. Measurements

Artificial pollination was conducted in accordance with the growth characteristics of various waxy maize varieties. To ensure the seed setting rate of the plant ear, each variety of waxy maize was pollinated on two occasions. After 14 (S1), 18 (S2), 22 (S3), and 26 (S4) days of pollination, field samples were collected and stored in liquid nitrogen freezing conditions. The time of artificial pollination and sample collection is shown in Appendix A.

#### 2.4.1. Carotenoid Content

The fresh waxy maize kernel (1.0000 g) was ground into a slurry using an ice bath, and an ethanol–acetone solution in a 1:1 volume ratio was added to extract for many times. The extract was constant volume to 25 mL volumetric flask, and the OD values at 663, 645 and 470 nm were determined. The carotenoid content was calculated according to the following equation:
$$C=8.73{OD}_{470}+2.11{OD}_{663}-9.360{OD}_{645}$$
$$Q=\frac{\left(C\times V\right)}{M}$$

*Q*: carotenoid content (mg/g); *C*: carotenoid concentration (mg/L); *V*: total extract (L); *M*: sample weight (g); *OD*_470_, *OD*_663_, and *OD*_645_ absorbance at 470 nm, 663 nm, and 645 nm, respectively.

#### 2.4.2. Soluble Sugar Content

The dry waxy maize kernel flour (0.1000 g) was accurately weighed into a 50 mL centrifuge tube, and then 4 mL of 80% ethyl alcohol was added. After heating in water bath 80 °C for 30 min, the mixture was centrifuged at 3000 rpm for 15 min. The process was repeated three times, the supernatant was combined, and activated carbon was added to a 50 mL volumetric flask. Then, 1 mL supernatant and 4 mL 0.2% anthrone solution were added to a new centrifuge tube. A UV–visible spectrophotometer set at 620 nm (Thermo Fisher Scientific Inc., Waltham, MA, USA) was used to measure the absorbance value at OD_620_ after the sample solution had cooled to room temperature after 15 min of boiling in a water bath. The regression equation was Y = 3.1223X + 0.0984 (R^2^ = 0.998), and the standard curve was created using the absorbance as the ordinate and the D-glucose and anhydrous amounts as the abscissa. All experimental treatments were biologically replicated three times.

#### 2.4.3. Vitamin C Content

After weighing the fresh waxy maize kernel flour (1.0000 g), a 25 mL volumetric flask was filled with a constant volume of oxalic acid EDTA solution. After centrifuging a part of the homogenate for 10 min at 3000 rpm, 10 milliliters of the supernatant were extracted. Then, 4 mL of a 5% ammonium molybdate solution, 2 mL of a 5% sulfuric acid solution, and 1 mL of a metaphosphoric acid–acetic acid solution were added. In a 25 mL volumetric flask containing distilled water, the volume remained steady. Additionally, a UV–visible spectrophotometer (Thermo Fisher Scientific Inc., Waltham, MA, USA) was used to assess the absorbance value at OD_705_ after 15 min. The standard curve was drawn with the vitamin C as the abscissa and the absorbance as the ordinate, and the regression equation was Y = 0.049X + 0.0018 (R^2^ = 0.998). All experimental treatments were biologically replicated three times.

#### 2.4.4. Anthocyanin Content

The fresh waxy maize kernel flour (3.0000 g) was weighed into the tube, and 20 mL 0.1 HCl–methanol solution was added. The extract was obtained via a 50 °C dark water bath for 1.5 h, with the process repeated twice, and the extract was combined. Then, the mixture was centrifuged at 6000 rpm for 10 min, and the supernatant was diluted to 100 mL. The absorbance value at OD_530_ was measured with a UV–visible spectrophotometer at 620 nm (Thermo Fisher Scientific Inc., Waltham, MA, USA). The standard curve was drawn with the cyanidin as the abscissa and the absorbance as the ordinate, and the regression equation was Y = 0.0126X − 0.0001 (R^2^ = 0.999). All experimental treatments were biologically replicated three times.

#### 2.4.5. Mineral Element Content

The dry waxy maize kernel flour (1.0000 g) was weighted into the desiccation pipe, and then 15 mL HNO_3_ and 10 mL H_2_O_2_ were added. An LWY84B alimentary furnace reaction system (Wuhan Gremo testing equipment Co., Ltd., Wuhan, China) was then used to digest them. The sample was diluted to 50 mL and filtered once it had cooled. Agilent 7700x inductively coupled plasma mass spectrometry (Agilent Technology Co., Ltd., Lexington, KY, USA) was used to examine the Fe, Mn, Zn, Cu, and Ca contents.

#### 2.4.6. Phenylalanine Content

The fresh waxy maize kernel flour was diluted with Na_2_HPO_4_–NaH_2_PO_4_ to 480–620 μg/mL. Then, 5 mL extract was transferred into the tube, and 1 mL ninhydrin was added. After heating in water bath 90 °C for 25 min, the absorbance value at OD_570_ was measured with a UV–visible spectrophotometer at 620 nm (Thermo Fisher Scientific Inc., Waltham, MA, USA). The standard curve was drawn with the β-phenylalanine as the abscissa and the absorbance as the ordinate, and the regression equation was Y = 0.0053X − 22.3992 (R^2^ = 0.997).

#### 2.4.7. Enzyme Activity in Anthocyanin Synthesis

The activities of phenylalanine ammonia lyase (PAL), chalcone isomerase (CHI), dihydroflavonol reductase (DFR), and flavonoid 3-glucosyltransferase (UFGT) were measured by ESISA kit from Shanghai Suolaibao Biotechnology Co., Ltd., Shanghai, China, [34].

### 2.5. Statistical Analysis

All data are presented as the mean ± standard error. Statistical analysis was performed using DPS 7.5, and the Origin 2024 was used for mapping. Duncan’s test was used to determine the significant differences in indicators among the different treatment reported at a significance level of *p* ≤ 0.05. The functional nutritional quality of maize was normalized for principal component analysis (PCA) in order to generate a correlation matrix. The correlation matrix was used to calculate the factor scores of the principal components and to determine eigenvalues and relative contribution rates. The Pearson method was used to perform correlation analysis, and the findings show varying degrees of correlation with corresponding probability values of *p* < 0.05, *p* < 0.01, and *p* < 0.001. Partial least squares (PLS) analysis was employed to model the relationships between precursor substance content and enzyme activity in anthocyanin synthesis and anthocyanin content.

## 3. Results

### 3.1. Functional Quality

Figure 1 shows the effects of varieties, kernel development stages, and their interactions on carotenoid content, soluble sugar content, vitamin C content, and anthocyanin content of maize, and they have significant effects.

With the delay in kernel development, carotenoid content of waxy maize showed a trend of first increasing and then decreasing, and reached the highest in S2. The yellow maize varieties (J41 and J7) had higher carotenoid content compared with white maize (J18 and W2000) and black maize varieties (J10 and J20). J41 had the highest carotenoid content, significantly higher than J18, W2000, J7, J10, and J20 by 41.27%, 77.73%, 31.61%, 49.68%, and 54.63% at S2 (Figure 1A).

Kernel development stage and variety had significant effects on soluble sugar content of waxy maize. The soluble sugar content was decreased with the delay in kernel development stage. The soluble sugar content of waxy maize did not change significantly with kernel color. J18 had the lowest soluble sugar content, which was significantly lower than W2000 and J10 by 34.82% and 34.33% at S1, and significantly lower than J7 and J10 by 30.34% and 40.07% at S4 (Figure 1B).

The vitamin C content of waxy maize was decreased first and then increased with the delay in kernel development stage. In addition, the vitamin C content of black maize varieties was higher than white and yellow maize varieties. At S1, S3, and S4, the vitamin C content of J10 and J20 were significantly higher than other varieties. At S2, W2000 had the highest vitamin C content, and significantly higher than J8, J41, J7, J10, and J20 by 48.83%, 35.25%, 41.84%, 41.21%, and 19.40%, respectively (Figure 1C).

The anthocyanin content of waxy maize was increased from S1 to S4. The black maize varieties had higher anthocyanin content compared with white and yellow maize varieties, and the anthocyanin content of with white and yellow maize had no significant difference. The anthocyanin content of J20 was significantly higher than J10 by 25.70%, 13.83%, and 14.88% at S1, S2, and S4. At S3, the anthocyanin content of J10 and J20 was significantly higher than J18, W2000, J41, and J7 by 204.67, 251.24, 251.15, 192.71 times and 202.24, 248.25, 248.16, and 190.42 times (Figure 1D).

### 3.2. Mineral Element

As shown in Figure 2, Fe, Mn, Cu, and Ca content of maize were significantly affected by varieties, kernel development stages, and their interactions, while the Zn content was significantly affected by kernel development stages and their interactions.

At S1 and S3, the waxy maize had higher Fe content, and it was lowest at S4. With the delay in kernel development stage, the Fe content of waxy maize decreased. Moreover, Fe content of waxy maize was significantly affected by varieties, but there was no significant correlation between Fe content and kernel color. J41, J20, W2000, and J10 had the highest Fe content at S1, S2, S3, and S4, respectively (Figure 2A–D).

As shown in Figure 2E–H, Mn content in waxy maize was significantly affected by kernel development stage and variety. From S1 to S4, Mn content of waxy maize showed a trend of first increasing and then decreasing, and reached the highest in S2. J7 had the lowest Mn content at S4, which was significantly lower than J18, W2000, J41, J10, and J20 by 57.39%, 56.87%, 60.94%, 57.08%, and 55.37%, respectively.

The response of Zn content in waxy maize kernels to kernel development stage was not significant. The Zn content of different varieties of waxy maize was significantly different, but there was no correlation with kernel color. J20 had the highest Zn content at S2, and J18 had the lowest Zn content, which was significantly lower than the highest by 44.45% (Figure 2I–L).

Both kernel development stage and variety had significant effects on the Cu content of waxy maize kernels. With the delay in kernel development stage, the Cu content of waxy maize showed a decreasing trend. In addition, the Cu content of black waxy maize was significantly higher than that of white and yellow. At S2 and S3, the Cu content of J20 was highest, and significantly higher than J18, W2000, J41, J7, and J10 by 80.39%, 74.50%, 66.31%, 65.39%, 59.22%, and 63.09%, 59.04%, 64.61%, 60.53%, 51.11% (Figure 2M–P).

As shown in Figure 2Q–T, Ca content in waxy maize was significantly affected by kernel development stage and variety. At S3, the Ca content of waxy maize was highest. Meanwhile, there was no correlation between Ca content and waxy maize with different kernel color. J18 had the highest Ca content at S3, which significantly higher than other varieties.

### 3.3. Principal Component Analysis

Principal component analysis (PCA) of different varieties of waxy maize at all four kernel development stages (Appendix A) and at S1 (Figure 3A), S2 (Figure 3B), S3 (Figure 3C), and S4 (Figure 3D) was conducted on the functional quality and mineral element. In the PCA of all four kernel development stages, there were three eigenvalues greater than one, and their contribution rates were 38.4%, 18.9%, and 15.0% (Appendix A). Soluble sugar, vitamin C, anthocyanin, Fe, Mn, Zn, Cu, and Ca content was positively correlated with PC1, the content of soluble sugar, vitamin C, Fe, Cu, and Ca was negatively correlated with PC2, while vitamin C and anthocyanin content was negatively correlated with PC3. The results show that with the delay in kernel development stage, the comprehensive quality of waxy maize showed a decreasing trend. Furthermore, the comprehensive quality scores of the six waxy maize varieties were as J10 > J20 > J41 > W2000 > J7 > J18 (Appendix A).

At S1, the PCA of quality for all waxy maize samples produced two statistically significant principal components accounting for 73.6% of the cumulative contribution to variance. The contributions of the first and second major components were 41.1% and 32.5%, respectively. In order to appropriately represent the variations in quality among various samples, PCA might be utilized. The score plot was used to separate the several types of waxy maize. The quality of the samples in nearby locations was comparable. J10 was situated on PC1 and PC2’s positive side, whereas J18 was situated on their negative side. The second quadrant has J7 and W2000, and the fourth quadrant contains J41 and J20. In contrast to J18, J10 showed higher levels of soluble sugar, Cu, Zn, Mn, vitamin C, and anthocyanin (Figure 3A).

As for S2, the first two principal components accounted for 71.5% of the data variance, and their contribution rates were 52.0%, and 119.5%, respectively. J10 and J20 were located on the positive side of PC1, indicating that these varieties had higher soluble sugar, vitamin C, Mn, anthocyanin, Zn, Cu, Ca, and Fe content. J7, J41, and W2000 are located in the second quadrant, which indicated that these varieties have a similar quality. Similarly, J18 was located in the third quadrant and exhibited a suboptimal quality (Figure 3B).

As shown in Figure 3C, the cumulative contribution rate of the first two principal components was 68.5% at S3 with PC1 (38.1%) and PC2 (30.4%). The six varieties are scattered in the score plot. J10, W2000, J41, and J7 were located on the positive side of PC1, while J20, J10, and W2000 were located on the positive side of PC2. J20 had higher Cu, vitamin C, and anthocyanin content, W2000 had higher soluble sugar, Zn, and Mn content. The content of Ca and Fe was higher in J18, and the carotenoid content was higher in J7 and J41.

At S4, the first principal component contribution was 50.0%, and the second was 32.5%. PC1 was positively correlated with the content of soluble sugar, Cu, Zn, Fe, Ca, and carotenoid and negatively correlated with anthocyanin, vitamin C, and Mn content. Similarly, PC1 was positively correlated with the Mn, vitamin C, anthocyanin, soluble sugar, Cu, Zn, and Fe content but negatively correlated with the content of Ca and carotenoid. W2000 and J18 as well as J7 and J41 were located on the same quadrant, indicating that these varieties had the same quality (Figure 3D).

### 3.4. Anthocyanin Synthesis

As shown in Figure 4A, both kernel development stage and variety had significant effects on the phenylalanine content of waxy maize kernels. The content of phenylalanine in waxy maize decreased with the delay in kernel development stage. At S1, the white waxy maize varieties had the highest phenylalanine content, followed by black waxy maize varieties and the phenylalanine content of white waxy maize varieties was lowest. At S2, J10 had the highest phenylalanine content, which was significantly higher than other varieties. As for S3, the phenylalanine content of J41 was significantly higher than J18, W2000, J7, J10, and J20 by 81.72%, 47.73%, 62.52%, 28.53%, and 78.44%, respectively. In addition, the phenylalanine content of J18, W2000, and J41 was higher than J7, J10, and J20 at S4.

With the delay in kernel development stage, PAL activity of white and yellow waxy maize varieties increased first and then decreased, and the PAL activity was highest in S2. In contrast, the black waxy maize variety J20 showed the opposite tendency. The PAL activity of black waxy maize varieties was significantly higher than that of yellow and white varieties. At S1 and S4, J20 had the highest PAL activity, while the PAL activity of J10 was highest at S2 and S3. At S4, the activity of PAL in J20 was significantly higher than J18, W2000, J41, J7, and J10 by 2.93, 2.40, 2.11, 2.38, and 1.18 times (Figure 4B).

From S1 to S4, the CHI activity of waxy maize increased. Varieties had significant effects on CHI activity, and black waxy maize varieties were significantly higher than white and yellow varieties. The CHI activity of J20 was significantly higher compared with J10 by 12.16% and 8.75% at S1 and S2. Whereas, there was no significant difference between J20 and J10 at S3 and S4 (Figure 4C).

Similarly, the delay in the kernel development stage resulted in an upward trend in the DFR activity of waxy maize. The DFR activity of black maize varieties (J10 and J20) was significantly higher than other varieties. At S1 and S4, the activity of DFR in J20 was highest, and significantly higher than J18, W2000, J41, J7, and J10 by 2.51, 2.06, 1.98, 1.88, 1.53 times and 2.13, 2.74, 2.18, 2.10, and 1.06 (Figure 4D).

As shown in Figure 4E, the UFGT activity of waxy maize kernels was significantly affected by variety and kernel development stage. The UFGT activity of waxy maize increased from S1 to S4. J10 and J20 had higher UFGT activity compared with J18, W2000, J41, and J7. J10 had the highest UFGT activity at S2 and S3, which was significantly higher by 2.09, 2.23, 1.60, 1.67, 1.11 times and 2.53, 1.74, 1.89, 1.62, 1.16 times.

### 3.5. Correlation Analysis

Correlation analysis was conducted to reveal the interrelationship among the anthocyanin content as well as enzymes and substances related to anthocyanin metabolism, and various significant correlations were found among these traits (Figure 5). PAL, CHI, DFR, and UFGT activity was positively correlated with anthocyanin content at S1, S2, S3, and S4. The phenylalanine content was negatively correlated with anthocyanin content at S1 and S4, while positively correlated with anthocyanin content at S2 and S3. Moreover, there were disparate degrees of correlation between four enzymes related to anthocyanin metabolism.

### 3.6. Regression Analysis

To reveal the relationships between enzymes and substances related to anthocyanin metabolism as well as the anthocyanin content, we conducted a PLS model (Figure 6). The result shows that PAL and CHI activity were the two most important indicators in deciding kernel anthocyanin content at S1. The three main indicators at S2 and S3 were the activity of PAL, CHI, and UFGT., whereas all four enzymes contributed significantly to anthocyanin synthesis at S4. From S1 to S4, the contribution of phenylalanine to the anthocyanin content gradually decreased. At the same time, the contribution of UFGT gradually increased. Obviously, PAL and CHI played a major role in the early stage of anthocyanin accumulation, while UFGT gradually took over in the middle and late stages.

## 4. Discussion

The PCA method enables for the extraction of independent factors from a large set of intercorrelated variables, thereby retaining trends and patterns while simplifying complex data [22]. Waxy maize has various health benefits, which is rich in functional substances such as carotenoid, soluble sugar, vitamin C, anthocyanin, and mineral element [11]. In this study, we comprehensively analyzed the functional quality of waxy maize with different kernel colors by PCA, and found that the comprehensive quality of waxy maize decreased with the delay in kernel development stage (Figure 3). The comprehensive quality scores of the six waxy maize varieties were as follows: J10, J20, J41, W2000, J7, and J18 (Appendix A). The black waxy maize varieties demonstrated superior functional quality. Suriano et al. [8] reported that the average carotenoid content was higher in yellow maize, followed by red maize, and lower levels were found in blue and purple maize. In this study, the carotenoid content of yellow waxy maize was the highest, which was significantly higher than that of white and black varieties. Obviously, the carotenoid content of yellow waxy maize varieties is higher than that of other varieties, which is also reflected in the color of the kernel. A similar conclusion was reached by Cai et al. [35], who proved that the carotenoids in yellow corn grains were dominated by lutein, followed by zeaxanthin, and low concentrations of β-carotene and β-cryptoxanthin. Yellow maize pigments can prevent cataracts and retinal macular degeneration by shielding the tissues of the eyes from blue light and monoclinic oxygen radicals [35]. A previous study showed that soluble sugar concentrations in sweet maize initially increased, peaked at 18–22 days after pollination, and then decreased [36]. However, our findings reveal a decline in the soluble content of waxy maize from 14 days after pollination. This may be due to the conversion of soluble sugars to starch during kernel filling [15]. In addition, the soluble sugar content of waxy maize with varying grain color was significantly different, yet no discernible correlation was observed with color. Sanahuja et al. [19] reported that the content of vitamin C was declined during kernel development, and similar results were also confirmed in our study. As the waxy corn kernels matured, the moisture content was reduced to a stable level and the vitamin C content was minimized and converted to an easily storable form [17]. Contrary to other functional substances, anthocyanin content increased with the development of maize kernels, indicating that anthocyanin is formed or deposited in the kernel during maturation [37]. The black waxy maize varieties (J10 and J20) had a higher anthocyanin content, which plays an important role in maize kernel coloration. Higher anthocyanin content can enhance antioxidant and biological activity, and is beneficial to anti-mutagenesis, anti-inflammatory, hypoglycemic, and antihypertensive activities [38]. In this study, the mineral element content of J41, J10, and J20 was higher compared with other varieties. It can be observed that the yellow and black waxy maize varieties exhibit a greater abundance of mineral elements [10].

The anthocyanin pigmentation requires the involvement of many enzymes and substances in the anthocyanin biosynthetic pathway [39]. For waxy maize to produce anthocyanin phenylalanine is the first constituent [25]. In this study, the phenylalanine content was negatively correlated with anthocyanin at S1 (14 DAP) and S4 (26 DAP), while being positively correlated with anthocyanin content 18–22 days after pollination (Figure 5). The contribution rate of phenylalanine to anthocyanin synthesis from S1–S4 was 0.273, 1.858, 0.161, and 0.021, respectively. The elevated phenylalanine content observed in colored maize (yellow and black) may be attributed to the increased demand for phenylalanine in anthocyanin synthesis [40]. In addition, PAL, CHI, DFR, and UFGT are the key enzymes in anthocyanin anabolic metabolism and the level of enzyme activity determines the anthocyanin content in different waxy maize varieties [26,27]. We found that in different kernel development stages, the PAL, CHI, DFR, and UFGT enzyme activity of black waxy maize was significantly higher than that of white and yellow waxy maize varieties. Obviously, the higher enzyme activity accelerated the anabolic metabolism of anthocyanin and increased the final anthocyanin content [26,27,39]. Furthermore, the activity of PAL, CHI, DFR, and UFGT was positively correlated with the anthocyanin content, which further proved that enzyme activity played an important role in anthocyanin anabolism. As the first key enzyme in anthocyanin synthesis, PAL activity demonstrated the most significant contribution to anthocyanin content in S3, indicating that PAL activity played a pivotal role in promoting anthocyanin synthesis at 22 days after pollination [28]. The greatest contribution to anthocyanin anabolic metabolism was observed in CHI at 14 days after pollination of waxy maize, after which it declined gradually. As for DFR activity, its contribution rate to anthocyanin was lower than that of other enzymes at four kernel development stages. In particular, during the S2, the activity of DFR has the least impact on anthocyanin content. Consequently, we postulated that DFR exerts a minimal influence on the rate of anthocyanin anabolism. With the development of the growth process, the contribution of UFGT activity to anthocyanin content increased gradually, and reached the highest value in S3 and the lowest value in S1. This indicates that the demand for UFGT for anthocyanin synthesis in the early stages of grouting is relatively low. Cai et al. [41] reported that the anthocyanins of dark fresh food maize were centaurin, pelargonidin, and peonidin. The content of total anthocyanins increased during the process of grain development. The content of total anthocyanins changed greatly in the early stage and slowed down in the later stage. The variation in anthocyanins in maize in different kernel development stages still needs further study.

## 5. Conclusions

Black waxy maize has high functional nutritional value, and the comprehensive quality scores of the six waxy maize varieties were as follows: J10, J20, J41, W2000, J7, and J18. The quality indicators of waxy maize exhibited disparate alterations across the various developmental stages, but its comprehensive functional quality decreased with the delay in kernel development stage. In addition, the phenylalanine content, as well as PAL, CHI, DFR, and UFGT activity, played an important role in anthocyanin biosynthetic pathway. The PAL and CHI played a major role in the early stage of anthocyanin accumulation, while UFGT gradually took over in the middle and late stages.

## Figures and Tables

**Figure 1 foods-14-00544-f001:**
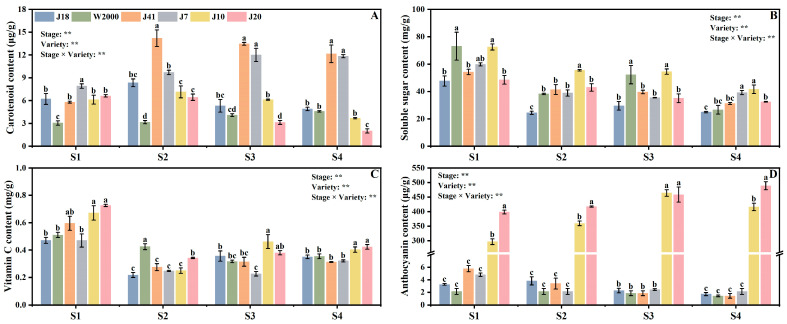
Functional quality of waxy maize with different kernel color in different kernel development stages. Functional quality indicators include carotenoid (**A**), soluble sugar (**B**), vitamin C (**C**), and anthocyanin (**D**). The means ± standard deviations are used to display the data. Values that do not display the same letter differ significantly (*p* < 0.05) among the varieties of the same stage. S1, S2, S3, and S4 means 14, 18, 22, and 26 days after pollination. ** indicate the significant differences at *p* ≤ 0.01.

**Figure 2 foods-14-00544-f002:**
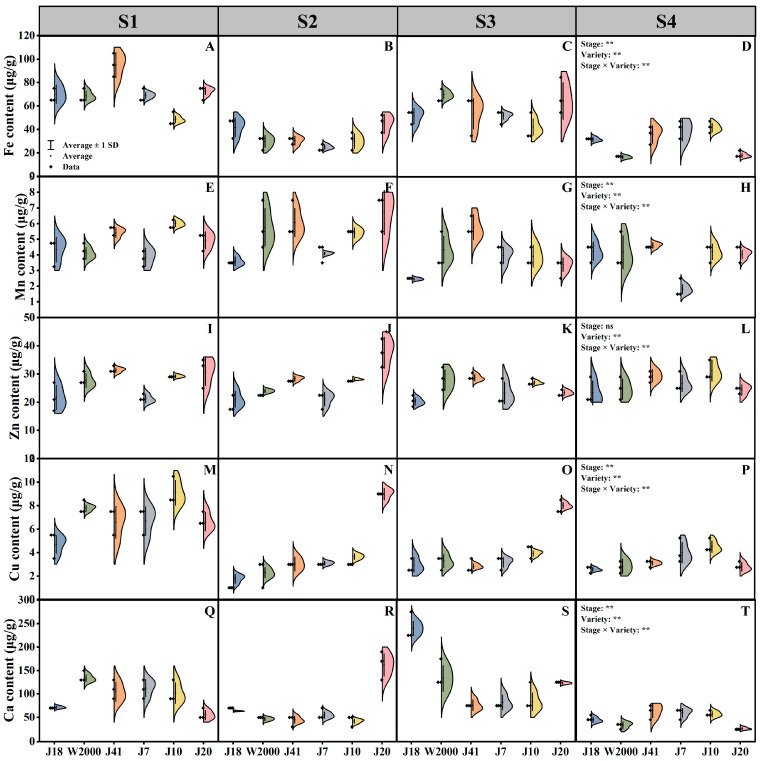
Mineral element of waxy maize with different kernel color in different kernel development stages. Mineral element indicators include Fe (**A**–**D**), Mn (**E**–**H**), Zn (**I**–**L**), Cu (**M**–**P**), and Ca (**Q**–**T**). Data are presented as means ± standard deviations. S1, S2, S3, and S4 means 14, 18, 22, and 26 days after pollination. ** indicate the significant differences at *p* ≤ 0.01.

**Figure 3 foods-14-00544-f003:**
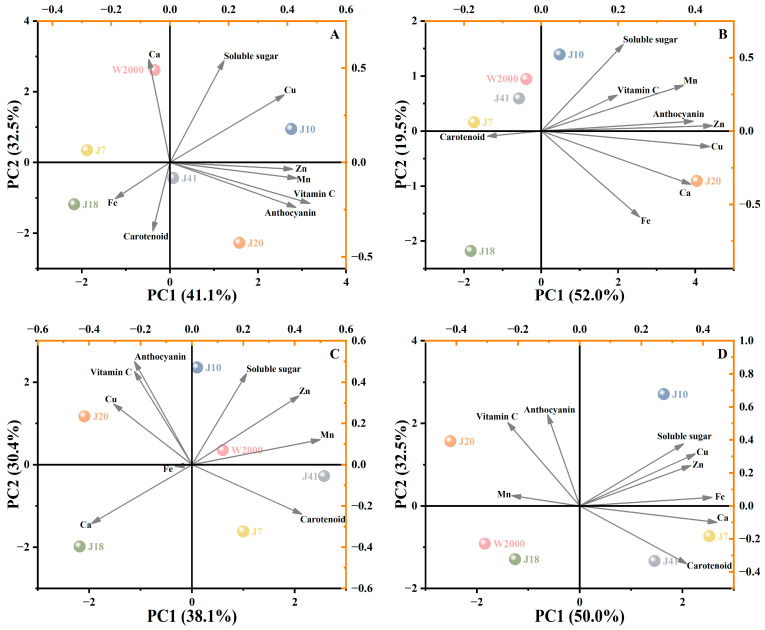
Principal component analysis of quality indicators for six waxy maize varieties at four kernel development stages. PCA load and score plot depicting the distribution of different varieties of waxy maize quality for the first two principal components at S1 (**A**), S2 (**B**), S3 (**C**), and S4 (**D**).

**Figure 4 foods-14-00544-f004:**
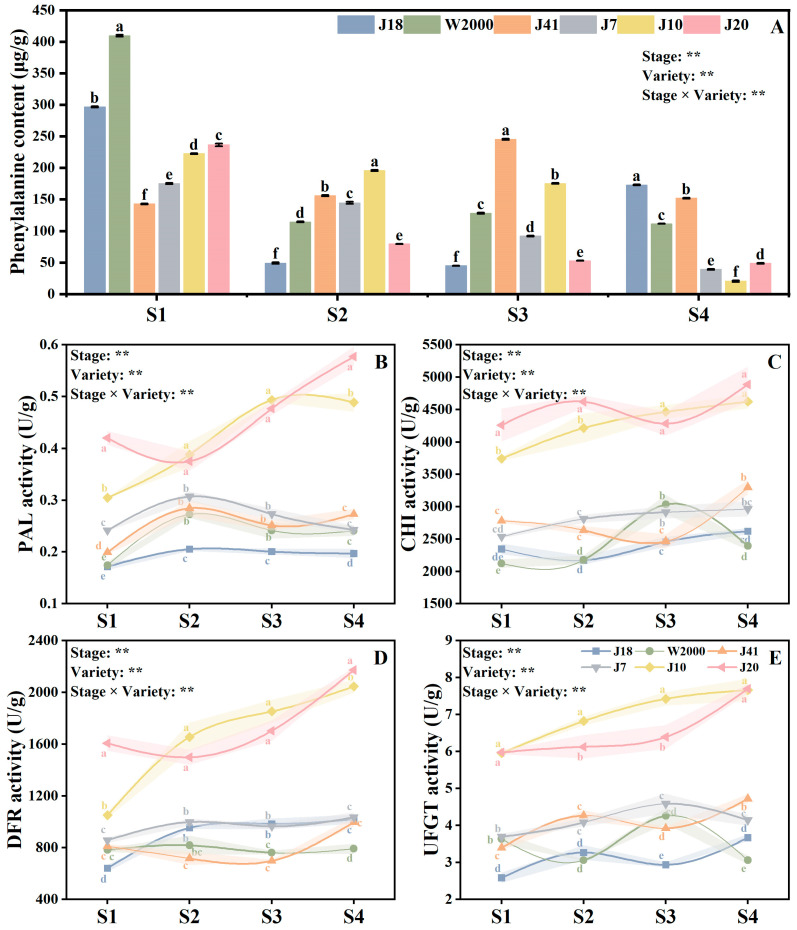
Precursor substance content and enzyme activity in anthocyanin synthesis of waxy maize with different kernel color in different kernel development stages. Precursor substance was anthocyanin (**A**). Enzyme activity in anthocyanin synthesis include PAL (**B**), CHI (**C**), DFR (**D**), and UFGT (**E**). The means ± standard deviations are used to display the data. Values that do not display the same letter differ significantly (*p* < 0.05) among the varieties of the same stage. S1, S2, S3, and S4 means 14, 18, 22, and 26 days after pollination. ** indicate the significant differences at *p* ≤ 0.01.

**Figure 5 foods-14-00544-f005:**
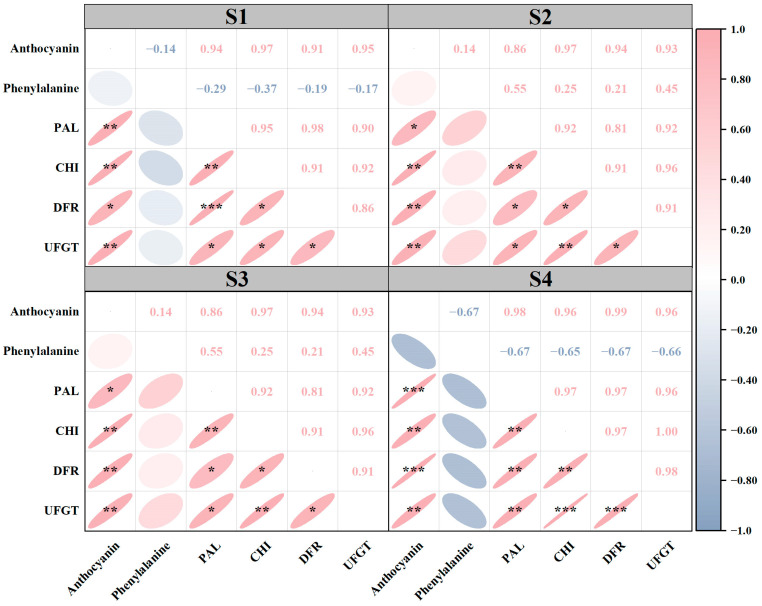
Correlation analysis of anthocyanin synthesis indicators for six waxy maize varieties at four kernel development stages. *, **, and *** indicate significance at 0.05, 0.01, and 0.001, respectively.

**Figure 6 foods-14-00544-f006:**
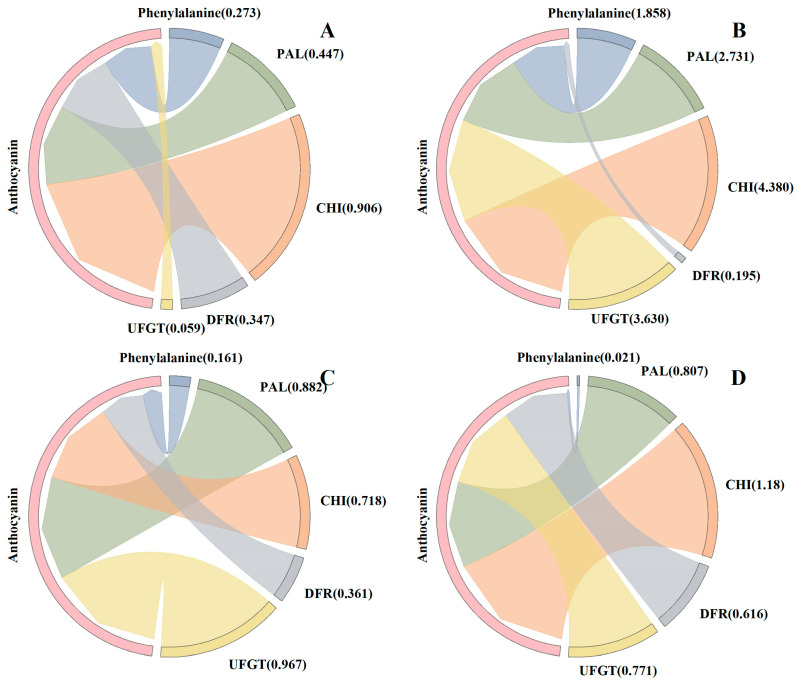
Regression coefficients (absolute values) for different indicators in the partial least squares (PLS) model at S1 (**A**), S2 (**B**), S3 (**C**), and S4 (**D**). Regression coefficients were calculated with standardized data. The regression coefficients represent the importance of different indicators in directly deciding the anthocyanin content of waxy maize in the PLS model.

## Data Availability

The original contributions presented in this study are included in the article/Appendix A. Further inquiries can be directed to the corresponding author.

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
