# Peer review of "Evaluation of Functional Quality of Maize with Different Grain Colors and Differences in Enzymatic Properties of Anthocyanin Metabolism"

_foods, 2025, doi:10.3390/foods14040544_

Round 1

Reviewer 1 Report

Comments and Suggestions for Authors

This manuscript reports some interesting data on the chemical composition of different waxy maize varieties. However, the authors should address many concerns to improve the current version of the manuscript.

The title of the manuscript, especially the second part, should be retitled to reflect the manuscript's content.

The Abstract, as a short version of the manuscript, must contain all relevant parts related to the research article. Add methodology, primary results, and conclusion. What does " rich in nutrition" mean?

Add some relevant data related to the global production and consumption of maize.

Lines 42-48; lines 53-55:  Avoid overemphasizing the importance without relevant scientific support. Instead, review the advantages and limitations related to the nutritional value of maize. Is there any data on the bioaccessibility of nutrients, especially minerals, from maize? Are there any antinutrients?

The Introduction should be divided into logical parts. If anthocyanin synthesis is the main topic of this article, it should be addressed from the beginning without providing distracting information.

Subtitle 2.1. It must be corrected.

Subtitle 2.4.6. Check the subtitle.

The results need to be adequately presented. The content presented in Figure 1 and Figure 2 is not visible. Explain what do S1, S2, S3 and S4 mean? The discussion needs to cover all presented results.

The quality of all Figures needs to be improved.

Author Response

As shown in attachment.

Reviewer 2 Report

Comments and Suggestions for Authors

In the present manuscript entitled "Evaluation of functional quality and regulation of anthocyanin biosynthesis metabolism of waxy maize with different kernel color” by Jing et al., authors have studied six waxy maize varieties of white, yellow and black to evaluate the changes of functional nutrients and the differences of anthocyanin anabolic pathways in maize kernels at different time of intervals after pollination. Overall, the manuscript is well written and the flow of the manuscript is so sound. It can be improved the quality of some figures:

Quality of the figures specially figure 1, 2, and 3 should be improved.  

Author Response

As shown in attachment.

Reviewer 3 Report

Comments and Suggestions for Authors

foods-3411341-peer-review-v1

Article 1

Evaluation of functional quality and regulation of anthocyanin biosynthesis metabolism of waxy maize with different kernel color

Suggestions:

1-Abstract: Unify as singular or plural word anthocyanin

2- Introduction

Avoid using etc.

There has been a surge in research due to the growing importance of anthocyanins in various sectors and their potential health benefits in recent years [22].

What sectors are you referring to? Please be more specific, improve your wording and mention them.

3- Statistical analysis

Mention the different analyses developed

4-- Materials and methods

4.1-In some determinations the following phrase is written

“All experimental treatments were biologically replicated three times.”

what does biologically replicated mean?

4.2- Mention how the results are presented in the tables

Include if appropriate “The means ± standard deviations are used to display the data”

5- Discussion

5.1-Improve the discussion regarding functional value, including recent references

5.2- Mention and include a discussion of bibliographic references, if possible, regarding the type of anthocyanins present in different types of corn, and at different stages of maturation.

The paper could be evaluated after the changes have been introduced, for possible acceptance.

Author Response

As shown in attachment.

Reviewer 4 Report

Comments and Suggestions for Authors

      I am reporting the review results on the manuscript entitled: "Evaluation of functional quality and regulation of anthocyanin biosynthesis metabolism of waxy maize with different kernel color". This study focuses on the functional properties and regulatory mechanisms governing anthocyanin production in waxy maize with various kernel colors. The research topic is of particular relevance because to the expanding market for nutritional foods and potential health advantages of anthocyanins. The manuscript integrates biochemical, enzymatic, and genetic important elements, resulting in a thorough comprehension of the researched features. Furthermore, the statistics are provided in a frequently understandable way making good use of figures and tables.

However, there are suggestions to author that will reveal an integral applicability to human health as:

-       The manuscript should cover the nutritional merits of the studied waxy maize varieties, including anti-inflammatory qualities, antioxidant potentiality, and the possible effect on human health.

-       The authors should go over the possible uses of these waxy maize variants in food products such as dietary/nutritional supplements, and beverages.

Comments on the Quality of English Language

 The manuscript is of appropriate English level

Author Response

As shown in attachment.

Round 2

Reviewer 1 Report

Comments and Suggestions for Authors

The authors made some changes in the revised version of the manuscript. However, their responses do not match the revised manuscript content.

The title of the manuscript does not reflect the manuscript content.

What does exploring anthocyanin biosynthesis metabolic regulation mean? 

Which factors influence anthocyanin biosynthesis?

Which tools and technologies did the authors use to analyze the metabolic regulations of anthocyanins?  

The Introduction is not in line with another part of the manuscript. The speculation on the health-prevention roles of maize in preventing any chronic disease without providing relevant supported data should be avoided.  What is the level of evidence that vitamin C or other nutrients prevent any diseases? This also should be addressed in the Discussion section.
The Figures and presentation results are not improved.

Overall, the results are valuable for publishing, but the manuscript needs to follow common scientific standards related to clarity, precision, structure, and objectivity.

Instead of further revising the current version, authors should prepare and resubmit a novel version for consideration.

Author Response

As shown in attachment.
